

# Stability of topological purity under random local unitaries

Salvatore F. E. Oliviero[1⋆], Lorenzo Leone[1], You Zhou[2,3,4] and Alioscia Hamma[1,5]

**1** Physics Department, University of Massachusetts Boston, 02125, USA
**2** Centre for Quantum Technologies. National University of Singapore,
3 Science Drive 2, Singapore 117543, Singapore
**3** Nanyang Quantum Hub, School of Physical and Mathematical Sciences,
Nanyang Technological University, Singapore 637371, Singapore
**4** Key Laboratory for Information Science of Electromagnetic Waves
(Ministry of Education), Fudan University, Shanghai 200433, China
**5** Université Grenoble Alpes, CNRS, LPMMC, 38000 Grenoble, France

⋆ s.oliviero001@umb.edu

## Abstract

In this work, we provide an analytical proof of the robustness of a form of topological entanglement under a model of random local perturbations. We define the notion of topological purity and show that, in the context of quantum double models, this quantity does detect topological order and is robust under the action of a random shallow quantum circuit.

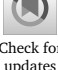

## Introduction

Topological order [1] is a novel kind of quantum order that goes beyond the paradigm of symmetry breaking. Its role is prominent in condensed matter theory as well as in quantum computation. In particular, topological order can be employed to construct various models for robust quantum memory and logic gates [2, 3]. Topologically ordered states show patterns of non local quantum entanglement that cannot be detected by a local order parameter. However, the long-range quantum entanglement leaves its mark in the reduced density matrix, and a series of papers have shown that topological order can be detected by the topological entropy [4–6]: a topological correction to the area law for the entanglement entropy. In particular, topological entanglement entropy has been employed to characterize the ground state of different models [7–17].

The presence of topological entanglement entropy is not identical with topological order: there are in fact spurious examples of topologically trivial states that nonetheless exhibit a non-zero topological entropy [18–21]. However, topological entanglement entropy is at least a very important probe of topological order. One important question is: if a topologically ordered state with non-zero topological entropy belongs to a gapped phase, is the topological entanglement entropy robust within that phase? In other words, if one perturbs the Hamiltonian whose ground state possesses topological order without closing a gap, will the topological entanglement entropy stay constant, or, at least, nonzero? There are rigorous proofs of the robustness of a quantum phase (i.e. the gap is not closing) that contain a topologically ordered state [22], but would that mean that the topological character of that state is preserved throughout the state, as revealed, for instance, by the topological entanglement entropy?

For specific forms of the perturbation, one can prove that the topological entanglement entropy is robust, see, [23] and [24]. From the numerical point of view, several results have shown such robustness under local perturbations of the Hamiltonian [25–28]. Other works have shown the robustness of topological entanglement entropy under small deformation of partition geometry. Quantum field theory arguments [5] suggest that topological entanglement entropy should always be robust within a phase, but an analytical proof for its robustness in lattice models is still lacking [29–31]. A remarkable result [32] exploits conditional independence of quantum states to prove robustness of topological entanglement entropy in $2D$ gapped system at the first order in perturbation theory; although the proof works fairly well for quantum double models, it is also limited to specific details of the perturbation.

In this work, we provide an analytic proof of the robustness of topological order under a noise model consisting of shallow circuit with random local unitaries. To this end, we construct a notion of topological subsystem purity that captures the same long-range pattern of entanglement of topological entanglement entropy, and we show that such topological purity is constant if the circuit is shallow compared to some relevant size of the subsystem.

We work in the framework of quantum double models on the cyclic group $\mathbb{Z}_d$ introduced by Kitaev in [2], and define the topological purity (TP), which is related to the topological $2-$Rényi entropy defined in [33]. There are many reasons to use purity instead of entanglement entropy in order to argue about questions about quantum many-body systems. Unlike the von Neumann entanglement entropy (whose measurement requires a complete state tomography of the system [34]), the $2-$Rényi entropy is directly related to the purity which is an observable and can be measured directly [35–39] as it is the expectation value of the swap operator over two copies of the system [40]. This quantity contains substantial information about the universal properties of quantum many-body systems [41] and it is able to reveal the topological pattern of entanglement [23, 33]. This property makes purity also amenable analytical treatment [42–47].

A phase of the matter is an ensemble of states that are considered equivalent: for example,

they enjoy the same symmetry (this criterion is not useful for topologically ordered states), or they are adiabatically connected, or they are connected by a shallow quantum circuit [48]. The kind of order that is revealed by this equivalence relation consists in the property that is conserved. For symmetry breaking states, the ordered phase is the phase of all the states that break the symmetry of the Hamiltonian in the same way (that is, they enjoy the same residual symmetry). In this context, the topological phase is revealed by the conservation of a topological pattern of entanglement.

To prove the robustness of topological order by the topological purity we introduce, as a noise model, a set of quantum maps whose action on a state is based on local random (shallow) quantum circuits. We find that the topological purity attains two different constant values in the two ensembles of states obtained by acting with the shallow quantum maps on two reference states, one topologically ordered and the other one topologically trivial, thereby defining two phases. When the circuit depth is comparable with the subsystem size, the long-range pattern of entanglement that is responsible for topological order can be changed and the topological purity can change value. The phase is then indeed the orbit of the so defined set of quantum maps through a reference state. The proof is obtained thanks to two key non trivial facts: (i) the subsystem purity (i.e. the purity of the reduced density matrix in a subsystem) of the ground state of $\mathbb{Z}_d$ quantum double models only depends on the geometry of the subsystem boundary, while the topological purity only depends on the subsystem topology, and (ii) the action of the specific noise model we work with can be regarded as the evolution of that boundary. Since the maps are shallow, their action will result in a local deformation of the subsystem boundary that does not alter their topology, and, by (i), this will result in an exactly constant topological purity. Similarly, we show that the topological purity of a topologically trivial state is zero and that it cannot be changed by our noise model.

The main idea of this work is the following. One defines subsystems $A, B, C$ such that the pair $(AB, BC)$ is in some sense topologically equivalent to the pair $(B, ABC)$. In a similar fashion as the topological entropy, the ratio of the purities

$$P_{top}(\sigma) := \frac{P_{AB}(\sigma)P_{BC}(\sigma)}{P_B(\sigma)P_{ABC}(\sigma)},$$

is non trivial ($< 1$) in a topologically ordered states $\sigma$ and detects the topological pattern of entanglement in such states. Then we consider a random quantum circuit $U_k = \prod_{i=1}^{k} U_{\tilde{X}_i}$ with $k$ gates acting on the qubits of the system. Every gate $U_{\tilde{X}_i}$ acts on the qubits in $\tilde{X}_i$. Now, after the action of $U_k$ the pure state of the system is mapped as $\sigma \to \sigma_k$ into another pure state. For every subsystem $\Lambda$, its purity will be $P_\Lambda(\sigma_k)$. By $\overline{(\ldots)}^k$ denote the average over the unitaries composing the circuit $U_k$. One can then define the quantity

$$\tilde{P} = \frac{\overline{P_{AB}(\sigma_k)}^k \overline{P_{BC}(\sigma_k)}^k}{\overline{P_B(\sigma_k)}^k \overline{P_{ABC}(\sigma_k)}^k}. \tag{1}$$

We will show that the above products (and ratios) of the average purities after randomizing over the unitaries in the circuit still show exactly the same topological pattern of entanglement, as long as the number $k$ of gates in the circuits is smaller than the relevant sizes of the system, namely the smallest of the length scales involved in the definition of the subsystems $A, B, C$. In order to make the proof, we show that the quantity $\tilde{P}$ can be seen as a functional on pairs of states. We will show that under the action of a unitary noise model based on shallow quantum circuits, the pattern of topological entanglement is preserved. Let us make a remark for the scope of clarity. In principle, one could define the topological purity by considering the ratios of the purities first, and then averaging over the unitaries in the quantum map. This quantity would be equivalent to the one we defined only if the ensemble fluctuations

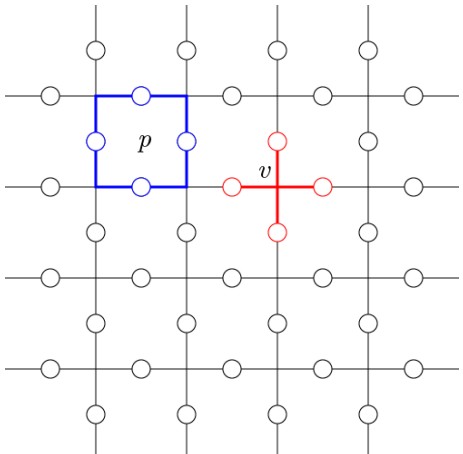

Figure 1: A system of spins on square lattice, plaquette and star are denoted respectively by $p$ and $v$.

were proven to be very small, which seems a formidable problem. Our definition goes around this problem because it nevertheless defines one quantity that has a constant value in the ensemble of states connected to the topologically ordered state and a different constant value in the trivial ensemble. The value of the definition is in its properties, that is, in being able to distinguish the two ensembles of states.

The paper is organized as follows: in Sec.1 we review $\mathbb{Z}_d$ quantum double models; in Sec.2 we introduce the notion of topological purity and discuss how it is connected with other measures of topological entropy; in Sec.3.1 we introduce the noise model and finally Secs.3.2 and 3.4, will be devoted to the rigorous proof of our result and will be rather technical.

# 1 Quantum Double models on $\mathbb{Z}_d$

Quantum double models are exactly solvable models defined on a lattice [2]. Consider the cyclic finite group $\mathbb{Z}_d$ with $|\mathbb{Z}_d| = d$ and local Hilbert spaces $\mathcal{H}_i \simeq \mathbb{C}^d$ and the total Hilbert space given by the tensor product of $N$ local Hilbert spaces, namely $\mathcal{H} = \bigotimes_{i=1}^N \mathcal{H}_i$ placed at the bonds of a square lattice $(V, E)$, see Fig.1. The dimension of the total Hilbert space is thus $D = d^N$. Without loss of generality for what concerns the calculation of topological entanglement entropy [49], we consider quantum double models on a torus. Let $B \equiv \{|n\rangle \, | \, n = 0, \ldots, d-1\}$ be an orthonormal basis in $\mathcal{H}_i \simeq \mathbb{C}^d$. For each local Hilbert space $\mathcal{H}_i$ we introduce the operators $\tilde{L}, \tilde{T}^{(j)}$ defined through their action on the ket $|n\rangle$:

$$\tilde{L}^m |n\rangle = |n + m\rangle \,, \quad \tilde{T}^{(m)} |n\rangle = \delta_{mn} |n\rangle \,, \tag{2}$$

where $\tilde{L}^m := \tilde{L}\tilde{L}\cdots\tilde{L}$ $m$ times and the addition is modulo $d$. Consider the enlarged operators $L_i := \tilde{L}_i \otimes \mathbb{1}_{\mathcal{H}\backslash i}$ and $T_i^{(m)} := \tilde{T}_i^{(m)} \otimes \mathbb{1}_{\mathcal{H}\backslash i}$ acting non trivially only on the site $i \in V$. Define the following operators acting non trivially on the subset $v \subset V$, sketched in Fig.1:

$$A_m(v) = \prod_{i \in v} L_i^m \,, \quad B(p) = \sum_{\substack{m_1, m_2, m_3, m_4 \\ m_1 + m_2 + m_3 + m_4 = 0 \mod d}} T_{i_1}^{(m_1)} T_{i_2}^{(m_2)} T_{i_3}^{(m_3)} T_{i_4}^{(m_4)} \,, \tag{3}$$

note that $B(p)$(plaquette operator) and $A(v) = d^{-1} \sum_{m=0}^{d-1} A_m(v)$(star operator) are projectors. At this point, the Hamiltonian of the quantum double model reads:

$$H_{QD} = \sum_v (\mathbb{1} - A(v)) + \sum_p (\mathbb{1} - B(p)), \tag{4}$$

and the ground state manifold $\mathcal{L}$ is given by:

$$\mathcal{L} = \{|\psi\rangle \in \mathcal{H} | A(v)|\psi\rangle = |\psi\rangle, B(p)|\psi\rangle = |\psi\rangle\}. \tag{5}$$

To represent the ground state in terms of the spin degrees of freedom, let us introduce $G$ the group generated by all the $A_m(v)$ operators, defined as $G = \langle\{A_m(v)|m = 0,\ldots,d-1, v = 1,\ldots,N/2\}\rangle$. The state $|\psi_{GS}\rangle$ defined as

$$|\psi_{GS}\rangle = \prod_s A(s)|0\rangle^{\otimes N} = d^{-N/2} \prod_s \sum_{m=0}^{d-1} A_m(s)|0\rangle^{\otimes N} = d^{-N/2} \sum_{h \in G} h|0\rangle^{\otimes N} \tag{6}$$

is a state in $\mathcal{L}$, as it can be readily checked. Other basis states in $\mathcal{L}$ can be constructed by the use of non contractible loop operators [2]. The topological order in this model can be detected by the entanglement entropy in the ground state manifold. Consider a bipartition in the Hilbert space, namely $\mathcal{H} = \mathcal{H}_\Lambda \otimes \mathcal{H}_{\bar{\Lambda}}$ and compute the reduced density matrix $\rho_\Lambda$[33]:

$$\rho_\Lambda = \mathrm{tr}_{\bar{\Lambda}} \Psi_0 = \frac{|G_{\bar{\Lambda}}|}{|G|} \sum_{h \in G/G_{\bar{\Lambda}}, \tilde{h} \in G_\Lambda} h_\Lambda^{-1}|0\rangle\langle 0|^{\otimes N} h_\Lambda \tilde{h}_\Lambda, \tag{7}$$

where $\Psi_0 \equiv |\psi_{GS}\rangle\langle\psi_{GS}|$ and we introduced $G_\Lambda := \{g \in G | g = g_\Lambda \otimes \mathbb{1}_{\bar{\Lambda}}\}$ and $G_{\bar{\Lambda}} := \{g \in G | g = \mathbb{1}_\Lambda \otimes g_{\bar{\Lambda}}\}$ that are normal groups in $G$, and the quotient groups $G/G_\Lambda$ and $G/G_{\bar{\Lambda}}$. Following [33] we can prove that $\rho_\Lambda^2 = \frac{|G_\Lambda||G_{\bar{\Lambda}}|}{|G|}\rho_\Lambda$ and thus the purity is given by $P_\Lambda(\rho) = \frac{|G_\Lambda||G_{\bar{\Lambda}}|}{|G|}$, i.e one can argue that the purity is counting the number of *independent* operators $A_m(v)$ acting non trivially on both regions $\Lambda$ and $\bar{\Lambda}$. Following [4], given a region $\Lambda$, the number of $A_m(v)$ operators acting on both subsystems $\Lambda$ and $\bar{\Lambda}$ is $d^{|\partial\Lambda|-n_2-2n_3}$ where $|\partial\Lambda|$ is the cardinality of the boundary of $\Lambda$, i.e the number of sites in $\bar{\Lambda}$ having at least one nearest neighbor inside $\Lambda$, and $n_i$, for $i = 2,3$, is the number of sites in $\bar{\Lambda}$ having $i$ nearest neighbors inside $\Lambda$. Thus $n_2 + 2n_3$ is a geometrical correction which depends on the shape of the region $\Lambda$. For example, if $\Lambda$ is a convex loop (a rectangle) $n_2 = n_3 = 0$. So far we accounted for the number of star operators acting on both subsystems, but not all of them are independent from each other because of the constraints on the ground state manifold in (5), in particular the condition $|\psi_{GS}\rangle \in \mathcal{L} \iff \prod_p B(p)|\psi_{GS}\rangle = |\psi_{GS}\rangle$. Following [4, 6] and defining $n_\partial(\Lambda)$ as the number of boundaries of $\Lambda$, we have that the number of independent star operators is $d^{|\partial\Lambda|-n_2-2n_3-n_\partial(\Lambda)}$, i.e for each boundary of $\Lambda$ we have that the number of independent star operators acting on both subsystems decreases of a factor scaling as $d^{-1}$. For $\Psi_{qd} \equiv |\Psi_{qd}\rangle\langle\Psi_{qd}|$ being the ground state of the quantum double model on $\mathbb{Z}_d$, we thus can finally write the following:

$$P_\Lambda(\Psi_{qd}) = 2^{-\log_2 d|\partial\Lambda|+\Gamma_\Lambda}, \tag{8}$$

where $\Gamma_\Lambda = \gamma_\Lambda + n_\partial(\Lambda)\gamma$ is the sum of a geometrical term $\gamma_\Lambda = \log_2 d(n_2 + 2n_3)$ which depends on the shape of the boundary $\partial\Lambda$ and a topological correction $n_\partial(\Lambda)\gamma$, due to the actual number of independent star operators, only related to the topology of $\Lambda$. This topological correction $\gamma \equiv \log_2 d$ is called *topological entropy*[4–6]. Eq.(8) is of fundamental importance for the reminder of the paper: it is telling us that the purity of the reduced density matrix in the ground state manifold of the topologically ordered quantum double model depends on the boundary $\partial\Lambda$ only.

## 2 The topological purity

In this section, we show how the topological pattern of entanglement involved in topologically ordered states [1] can be also found in a new quantity: the topological purity (TP).

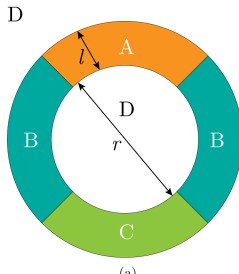 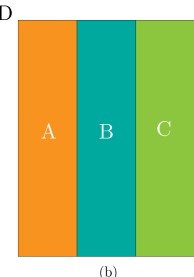

Figure 2: $(a)$ The graph configuration to define the topological entropy, i.e. $I(A;C|B)$. $l$ is the feature size of this graph configuration and $l \sim O(N)$, where $N^2$ is the total number of qudits, while $r \sim O(N)$ is radius of this topologically non trivial domain. Here and throughout the paper we assume $r > l$. $(b)$ $I(A;C|B) = 0$ in this simple graph configuration, even though there is long-range entanglement of the ground state.

To understand heuristically how this quantity works, let us first introduce the topological entropy. Consider the state $\sigma$ living in the Hilbert space $\mathcal{H} \equiv \mathcal{H}_A \otimes \mathcal{H}_B \otimes \mathcal{H}_C \otimes \mathcal{H}_D$ and the regions $AB, BC, B$ and $ABC$ drawn in Fig.2 $(a)$. The topological entropy is defined as

$$S_{top}(\sigma) = S_{ABC}(\sigma) + S_B(\sigma) - S_{AB}(\sigma) - S_{BC}(\sigma) \equiv -I(A;C|B), \tag{9}$$

where $S_\Lambda(\sigma)$ labels the von Neumann entropy of $\mathrm{tr}_{\bar{\Lambda}}(\sigma)$ where $\bar{\Lambda}$ is the complement of $\Lambda$ with respect to $ABCD$. The von Neumann entropy has several important theoretical properties, specifically strong subadditivity [50]. In the context of topological entanglement, this results in $I(A;C|B) \geq 0$. On the other hand, the $2-$Rényi entropy is also a good measure of entanglement and it has the advantage of being more amenable for our scope. Moreover, this quantity can be measured without resorting to complete state tomography [37–39].

As it was shown in [33], also all the Rényi topological entropies give exactly the same results for quantum double models. The definition of the topological entropy is equal to minus the quantum conditional information $I(A;C|B)$[50], which is an entropy quantity describing tripartite correlations of quantum states.

On the other hand, the purity of the state $\sigma$ in the subsystem $\Lambda$ is defined as $P_\Lambda(\sigma) := \mathrm{tr}\left[(\mathrm{tr}_{\bar{\Lambda}}\sigma)^2\right]$, where $\bar{\Lambda}$ is the complement of $\Lambda$.

In the same fashion of Eq.(9), we define the quantity

$$P_{top}(\sigma) := \frac{P_{AB}(\sigma)P_{BC}(\sigma)}{P_B(\sigma)P_{ABC}(\sigma)}, \tag{10}$$

i.e. the ratio of purities of the reduced density matrix of a quantum state $\sigma$ for the four subsystems $AB$, $BC$, $B$ and $ABC$. By definition,

$$-\log P_{top}(\sigma) = -\log P_{AB}(\sigma) - \log P_{BC}(\sigma) + \log P_B(\sigma) + \log P_{ABC}(\sigma) \tag{11}$$

is just the topological 2-Rényi entropy. A topologically ordered state will feature $P_{top}(\sigma) < 1$ while for a topologically trivial state $P_{top}(\sigma) = 1$. In other words, the product (ratios) of purities in $P_{top}$ show the topological entanglement pattern of a topologically ordered state.

Consider the domain $ABC$ in Fig.2 $(a)$, and the ground state of a quantum double model $\Psi_{qd}$ for which $P_\Lambda(\Psi_{qd}) = 2^{-\log_2 d|\partial\Lambda| + \Gamma_\Lambda}$ for each $\Lambda \in \{AB, BC, B, ABC\}$ and since $|\partial AB| + |\partial BC| = |\partial B| + |\partial ABC|$, $P_{top}$ is just given by the sum of the geometrical corrections

$$P_{top}(\Psi_{qd}) = 2^{\Gamma_{AB} + \Gamma_{BC} - \Gamma_B - \Gamma_{ABC}} \equiv 2^{-2\gamma}, \tag{12}$$

where $-2\gamma \equiv \Gamma_{AB} + \Gamma_{BC} - \Gamma_B - \Gamma_{ABC}$ is the topological entropy [6]. Note that, according to the discussion in the previous section, all the geometrical corrections related to the shape of the boundary $\partial(ABC)$ are canceled by the choice of the partitions $AB, BC, B, ABC$, namely $\gamma_{AB} + \gamma_{BC} = \gamma_B + \gamma_{ABC}$, and the only surviving term is the topological correction that does not depend on the shape of the boundary: it is a purely topological correction $\propto \gamma$. This correction is the mark of the topological phase. It is worth noting that the topological correction would not be detected from $P_{top}$ if $ABC$ was a simply connected region as the one sketched in Fig.2, see also [6]. That is because, as shown in the previous section, the number of boundaries $n_\partial(\Lambda)$ gives the number of topological corrections $\gamma$ to the purity $P_\Lambda$ of the related subsystem $\Lambda$. Specifically, consider Fig.2 $(b)$ first: we have $n_\partial(AB) = n_\partial(BC) = n_\partial(B) = n_\partial(ABC) = 1$, and thus according to Eq.(12) we have $2\gamma - 2\gamma = 0$, while for Fig.2 $(a)$: $n_\partial(AB) = n_\partial(BC) = 1$ and $n_\partial(B) = n_\partial(ABC) = 2$, thus $2\gamma - 4\gamma = -2\gamma$.

We now exploit a standard trick based on the swap operator to express $P_{top}$ in terms of expectation values. Let $\mathcal{H}_V \simeq \mathbb{C}^D \simeq \mathbb{C}^{d \otimes N}$ be the $D-$dimensional Hilbert space of $N$ qudits in a set $V$. Here, the Hilbert space of the $x-$th qudit is denoted by $\mathcal{H}_x \simeq \mathbb{C}^d$. Let $\Lambda \subset V$ be a subset of these qudits and $\mathcal{H}_\Lambda = \otimes_{x \in \Lambda} \mathcal{H}_x$ the corresponding Hilbert space. Let $\tilde{T}_\Lambda$ be the order two permutation (swap) operator on $\mathcal{H}_\Lambda^{\otimes 2}$ and let $T_\Lambda = \tilde{T}_\Lambda \otimes \mathbb{1}_{\bar\Lambda}$ be its trivial completion on the full $\mathcal{H}_\Lambda^{\otimes 2} \otimes \mathcal{H}_{\bar\Lambda}^{\otimes 2}$.

The purity of the state $\sigma$ in the bipartition $\mathcal{H}_\Lambda \otimes \mathcal{H}_{\bar\Lambda}$ is given by

$$P_\Lambda(\sigma) \equiv \operatorname{tr}_\Lambda \sigma_\Lambda^2 = \operatorname{tr}(\sigma^{\otimes 2} T_\Lambda) \equiv \langle T_\Lambda \rangle_{\sigma^{\otimes 2}}, \tag{13}$$

where $\sigma_\Lambda := \operatorname{tr}_{\bar\Lambda} \sigma$. The above chain of relations is telling us that the purity is from both the analytical point of view and the experimental point of view a quantity defined on two copies of the Hilbert space $\mathcal{H}$. In practice, in order to measure the purity of a quantum state $\sigma$ in a given bipartition $\mathcal{H}_\Lambda \otimes \mathcal{H}_{\bar\Lambda}$, one needs three steps: $(i)$ to prepare two identical copies of $\sigma$, $(ii)$ to build the observable *swap operator* on the subspace $\Lambda$ and finally, $(iii)$ to take the quantum expectation value of $T_\Lambda$ in $\sigma \otimes \sigma$, namely $\langle T_\Lambda \rangle_{\sigma^{\otimes 2}}$. Similarly, in order to measure $P_{top}$ defined in Eq.(10), one needs to repeat the steps $(i)$, $(ii)$ and $(iii)$ for the four observables $T_{AB}, T_{BC}, T_B, T_{ABC}$ and then combine them in the following way:

$$P_{top}(\sigma) = \frac{\langle T_{AB} \rangle_{\sigma^{\otimes 2}} \langle T_{BC} \rangle_{\sigma^{\otimes 2}}}{\langle T_B \rangle_{\sigma^{\otimes 2}} \langle T_{ABC} \rangle_{\sigma^{\otimes 2}}}. \tag{14}$$

The quantity $P_\Lambda(\sigma)$ is the purity of $\sigma$ when as a linear functional over $\sigma^{\otimes 2}$, that is, product states $\Psi^{\otimes 2}$ of $\mathcal{H}^{\otimes 2}$. We now extend this definition to arbitrary states $\Psi \in \mathcal{H}^{\otimes 2}$. We define *Topological purity* (TP) the quantity

$$\tilde{P}_{top}(\Psi) := \frac{\langle T_{AB} \rangle_\Psi \langle T_{BC} \rangle_\Psi}{\langle T_B \rangle_\Psi \langle T_{ABC} \rangle_\Psi}. \tag{15}$$

Obviously, for product states $\Psi^{\otimes 2}$ of $\mathcal{H}^{\otimes 2}$ one has $\tilde{P}_{top}(\Psi) = P_{top}(\Psi)$. With this result, if $\Psi_{qd}$ is the topologically ordered ground state of a quantum double model, its topological purity $\tilde{P}_{top}(\Psi_{qd})$ will be

$$\tilde{P}_{top}(\Psi_{qd}) = 2^{-2\gamma}. \tag{16}$$

On the other hand, if $\Psi_{triv}$ is a state belonging to a topologically trivial phase with no topological entanglement entropy, it will also be true that

$$\tilde{P}_{top}(\Psi_{triv}) = 1. \tag{17}$$

We have therefore established that the topological purity distinguishes these two states. In the next section, we show how this new definition helps us to prove that this quantity is robust

under a quantum map based on a shallow quantum circuit. This protocol to detect topological order under the noisy channel we defined is experimentally realizable on a quantum processor [51] using the techniques based on randomized measurements [37, 40].

# 3 Stability of topological purity

In this section, we establish a noise model based on quenched disorder, and show how the topological purity behaves under the noise model.

The noise model consists in a quantum channel $\mathcal{R}_U$ based on (shallow) random quantum circuits $U_k$. The quantum channel has as an input two copies of the initially topologically ordered state $\Psi$:

$$\mathcal{R}_{U_k} : \Psi^{\otimes 2} \mapsto \mathcal{R}_{U_k}(\Psi^{\otimes 2}). \tag{18}$$

In the above, $U_k$ is a random quantum circuit with $k$ gates. The gates act on a subset of the qubits on the graph $\Lambda$, that is, $\tilde{X}_i \subset \Lambda$ for $i = 1, \dots, k$. The random quantum circuit has thus the form

$$U_k = \prod_{i=1}^{k} U_{\tilde{X}_i}. \tag{19}$$

We say the map is based on quenched disorder because it acts as

$$\mathcal{R}_{U_k}(\Psi^{\otimes 2}) := \int d\mu(U|\tilde{X}_1) \dots d\mu(U|\tilde{X}_k) U_k^{\otimes 2} \Psi^{\otimes 2} U_k^{\dagger \otimes 2}. \tag{20}$$

Notice that the sequence $S = (\tilde{X}_k, \dots, \tilde{X}_1)$ completely characterizes the map. For this reason, we will also denote the above quantum channel by $\mathcal{R}_S$ when we want to make explicit the dependency on the sequence $S$. Operationally, this quantum channel maps the input state in a mixed state obtained by collecting several outputs of the random quantum circuit. If the sampling is good enough, the output state has the form Eq.(20). The output of the channel is now a mixed, non-separable state in $\mathcal{H}^{\otimes 2}$ for which the topological purity Eq.(15) is well defined. From the experimental point of view, this is the purity one would measure in an experiment if the measurement time-scales are much longer than the random fluctuations in the unitary noise.

The reason why the quantum channel $\mathcal{R}_{U_k}$ allows us to prove the stability of topological purity is that the evolution of $\Psi^{\otimes 2}$ under $\mathcal{R}_{U_k}$ can be mapped - for the sake of computing subsystem purities - in the evolution of the boundary of the subsystem, which we show in the next subsection. Then, we show that such boundary evolution exactly preserves $\tilde{P}_{top}$ provided that the number of gates $k$ is smaller compared to the smallest length scale in the subsystems $A, B, C, D$, see Fig. 2.

## 3.1 Topological purity and phases

Let us now dive into the technical details of the noise model. Since the purity is defined on $\mathcal{H}^{\otimes 2}$, we define a noise model on states living on two copies of the Hilbert space in the following way: let $X \subset V$ be a set of qudits with Hilbert space $\mathcal{H}_X := \otimes_{x \in X} \mathcal{H}_x$ and $d_X := \dim \mathcal{H}_X$. Let $U_X$ be a local unitary operator operating on the region $X$, i.e. operating on all the qubits contained in $X$. Let $U_X^{\otimes 2}$ be two copies of $U_X$, then after operating on $\sigma^{\otimes 2}$ with the unitary $U_X^{\otimes 2}$, we have $\sigma^{\otimes 2} \mapsto U_X^{\otimes 2} \sigma^{\otimes 2} U_X^{\dagger \otimes 2}$ and the purity becomes

$$\langle T_\Lambda \rangle_{\sigma^{\otimes 2}} \mapsto \langle T_\Lambda \rangle_{U_X^{\otimes 2} \sigma^{\otimes 2} U_X^{\dagger \otimes 2}} \equiv \text{tr}\left(T_\Lambda U_X^{\otimes 2} \sigma^{\otimes 2} U_X^{\dagger \otimes 2}\right). \tag{21}$$

We now choose $U_X$ to be a random unitary operator and define the following quantum map acting on $\sigma^{\otimes 2}$:

$$R_X(\sigma^{\otimes 2}) := \int d\mu(U|X)(U_X)^{\otimes 2}\sigma^{\otimes 2}(U_X)^{\dagger \otimes 2}, \tag{22}$$

where $d\mu(U|X)$ is the Haar measure over the unitary group $\mathcal{U}(\mathcal{H}_X)$. Therefore, fixed $X \subset V$, the map $R_X(\cdot)$ randomizes over the action of the full unitary group on $\mathcal{H}_X^{\otimes 2}$. Thus, after the noise on $X$, the purity becomes:

$$\langle T_\Lambda \rangle_{\sigma^{\otimes 2}} \mapsto \langle T_\Lambda \rangle_{R_X(\sigma^{\otimes 2})} \equiv \mathrm{tr}\left(T_\Lambda R_X(\sigma^{\otimes 2})\right), \tag{23}$$

note that the above operation is no more acting independently on the single copies of $\mathcal{H}$, but it is entangling them in $\mathcal{H}^{\otimes 2}$. So far this is a single $X$ noise model. In order to generalize it to more than one single $X$ domain, consider an ordered string of subsets $S = \{\tilde{X}_1, \ldots, \tilde{X}_k\}$ and random unitary operators $U_{\tilde{X}_i}^{\otimes 2}, i = 1, \ldots, k$ operating on the corresponding subset $\tilde{X}_i$ and acting on $\sigma^{\otimes 2}$ in an ordered way, namely $\sigma^{\otimes 2} \mapsto U_{\tilde{X}_k}^{\otimes 2} \cdots U_{\tilde{X}_1}^{\otimes 2}\sigma^{\otimes 2}U_{\tilde{X}_1}^{\dagger \otimes 2} \cdots U_{\tilde{X}_k}^{\dagger \otimes 2}$. We define the quantum map randomizing over the action of these gates as:

$$\mathcal{R}_S(\sigma^{\otimes 2}) := R_{\tilde{X}_k} \cdots R_{\tilde{X}_1}(\sigma^{\otimes 2}), \tag{24}$$

where

$$R_{\tilde{X}_i} : \mathcal{O} \mapsto R_{X_i}(\mathcal{O}) := \int d\mu(U|\tilde{X}_i)(U_{\tilde{X}_i})^{\otimes 2}\mathcal{O}(U_{\tilde{X}_i})^{\dagger \otimes 2}, \quad \mathcal{O} \in \mathcal{B}(\mathcal{H}^{\otimes 2}). \tag{25}$$

As we remarked above, the string $S$ completely characterizes the map. For each string of domains $S$, the action of $\mathcal{R}_S$ on a state of $\mathcal{H}^{\otimes 2}$ describes the average action of a given random quantum circuit operating in the region $\tilde{X}_i \in S$, therefore at this point we define the set $\mathcal{S}$ of all such strings:

$$\mathcal{S} := \{S = \{\tilde{X}_1, \ldots, \tilde{X}_k | \tilde{X}_i \subset V, i = 1, \ldots, k\}, k \in \mathbb{N}\}. \tag{26}$$

It is straightforward to see that, for any subset $\tilde{\mathcal{S}} \subset \mathcal{S}$, the action of $\mathcal{R}_S$ on a state $\sigma^{\otimes 2}$ varying $S \in \tilde{\mathcal{S}}$ creates an ensemble of states living on $\mathcal{H}^{\otimes 2}$ as follows:

$$\mathcal{E}_{\tilde{\mathcal{S}}}(\sigma^{\otimes 2}) := \{\mathcal{R}_S(\sigma^{\otimes 2}) \in \mathcal{B}(\mathcal{H}^{\otimes 2}) | S \in \tilde{\mathcal{S}}\}, \tag{27}$$

i.e. the ensemble of states $\mathcal{E}_{\tilde{\mathcal{S}}}(\sigma^{\otimes 2})$ contains all the states $\Psi_S$ living in $\mathcal{H}^{\otimes 2}$ obtained by the action of $\mathcal{R}_S$ varying $S$ in a subset $\tilde{\mathcal{S}}$ of $\mathcal{S}$, defined in Eq.(26). Notice that each string of ordered domains $S$ describes a quantum circuit consisting of random gates with support on $\tilde{X}_i \in S$. In the following definition, we give the notion of $l-$ shallow string, which is produced by the action of a shallow quantum circuit (although the opposite is not true, i.e. a shallow string can be also generated by a non-shallow circuit):

**Definition 1** ($l-$shallow string). *Let $S$ be a string of domains. $S = \{X_1, \ldots, X_k | X_i \subset V\}$ is an $l-$shallow string iff* $\mathrm{diam}(X_1 \cup \cdots \cup X_k) < l$.

Now we can enunciate the main result of this paper: in the following theorem we prove that the topological purity attains a constant value in the ensembles of states obtained from both the ground state of the quantum double model on $\mathbb{Z}_d$ $\Psi_{qd}$ and a topologically trivial pure state $\Psi_{triv}$, provided that the subset $\tilde{\mathcal{S}} \subset \mathcal{S}$ contains strings of domains $S$ describing shallow quantum circuits. Since this definition contains circuits with trivial action, this value is also the value of the topological purity in the initial state.

**Theorem.** *Let $\Psi_{qd}$ be the ground state of a quantum double model and let $\Psi_{triv}$ be a pure, topologically trivial quantum state. Let $\tilde{\mathcal{S}}_l \subset \mathcal{S}$ be the subset of all possible $l-$shallow strings defined in Definition 1, then the topological purity is constant in the following ensembles of states: $\mathcal{E}_{\tilde{\mathcal{S}}_l}(\Psi_0^{\otimes 2})$, $\mathcal{E}_{\tilde{\mathcal{S}}_l}(\Phi^{\otimes 2})$, namely:*

$$\tilde{P}_{top}(\Psi_S) = 2^{-2\gamma}, \quad \forall \Psi_S \in \mathcal{E}_{\tilde{\mathcal{S}}_l}(\Psi_{qd}^{\otimes 2}), \tag{28}$$

*and*

$$\tilde{P}_{top}(\Phi_S) = 1, \quad \forall \Phi_S \in \mathcal{E}_{\tilde{\mathcal{S}}_l}(\Psi_{triv}^{\otimes 2}), \tag{29}$$

morally an $l-$shallow string $S \in \tilde{\mathcal{S}}_l$ describes a shallow random quantum circuit that do not destroy the topological nature of $\Psi_{qd}^{\otimes 2}$. Since we found that the TP gets a constant value in two distinct ensemble of states, we claim that the topological purity is stable in the topological ordered phase $\mathcal{E}_{\tilde{S}}(\Psi_{qd}^{\otimes 2})$ and in the topological trivial phase $\mathcal{E}_{\tilde{S}}(\Psi_{triv}^{\otimes 2})$. Based on Fig. 2, we remark that an $l-$shallow string is an $r-$shallow string as well, being $r > l$.

The proof of this theorem is in Sec.3.4. As we stated at the beginning of this section, the proof descends from two facts:

(i) the *purity dynamics* generated by $\mathcal{R}$ purity averaged over the noise is equal results in a boundary evolution for the subsystem. The purity of the output state is equal to the purity of the initial state for the subsystem corresponding to the evolved boundary.

(ii) For a shallow map, the boundary evolves in a way that the topological purity stays exactly constant.

We start proving the fact (i) in the next subsection. Subsection 3.3 will instead show fact (ii).

## 3.2 Purity dynamics under random quantum circuits

In this section, we show how, under the noise model defined by the quantum map Eq.(22), the evolution of the purity becomes a boundary evolution for the purity in the initial state.

Consider a state $\sigma$ and the swap operator $T_\Lambda$ defined in the region $\Lambda$. As shown in Sec. 3.1 the purity of $\sigma$ in the region $\Lambda$ is the expectation value of the swap operator $T_\Lambda$ computed on two copies of the state $\sigma$, $P_\Lambda(\sigma) \equiv \langle T_\Lambda \rangle_{\sigma^{\otimes 2}}$. Let $R_X$ be the quantum map defined in Eq.(22) and consider the expectation value of $T_\Lambda$ in the state $R_X(\sigma^{\otimes 2})$; because $R_X(\cdot)$ is an hermitian and self-dual operator [52], we can equivalently write:

$$\langle T_\Lambda \rangle_{R_X(\sigma^{\otimes 2})} = \langle R_X(T_\Lambda) \rangle_{\sigma^{\otimes 2}}, \tag{30}$$

i.e. the expectation value of the swap operator $T_\Lambda$ on the state $R_X(\sigma^{\otimes 2})$ is equal to the expectation value of the evolved swap operator $R_X(T_\Lambda)$, i.e. the image of $T_\Lambda$ under the map $R_X(\cdot)$, on the original state $\sigma^{\otimes 2}$. In practice, we are considering the Heisenberg picture for the evolution of the swap operator. This point of view is convenient for us, because - thanks to the simple equation (30) - the purity dynamics can be described as the dynamics of the boundary $\partial \Lambda$ of the region $\Lambda$ and thus can be described in terms of patching. First of all, let $X \subset V$ be a domain and let us compute the action of the map $R_X(\cdot)$ on the swap $T_\Lambda$. One can show [52]:

$$R_X(T_\Lambda) = \begin{cases} N_{d_{\Lambda\backslash X}} T_{\Lambda\backslash X} + N_{d_{\Lambda\cup X}} T_{\Lambda\cup X}, & X \cap \partial\Lambda \neq \emptyset, X \cap \partial\bar{\Lambda} \neq \emptyset, \\ T_\Lambda, & X \cap \partial\Lambda = \emptyset, \end{cases} \tag{31}$$

where $N_{d_{\Lambda\backslash X}} := (d_X^2 - d_{\Lambda\cap X}^2)d_{\Lambda\cap X}^{-1}/(d_X^2 - 1)$ and $N_{d_{\Lambda\cup X}} := d_X(d_{\Lambda\cap X}^2 - 1)d_{\Lambda\cap X}^{-1}/(d_X^2 - 1)$ and $d_{\Lambda\cap X} = \dim \mathcal{H}_{\Lambda\cap X}$.

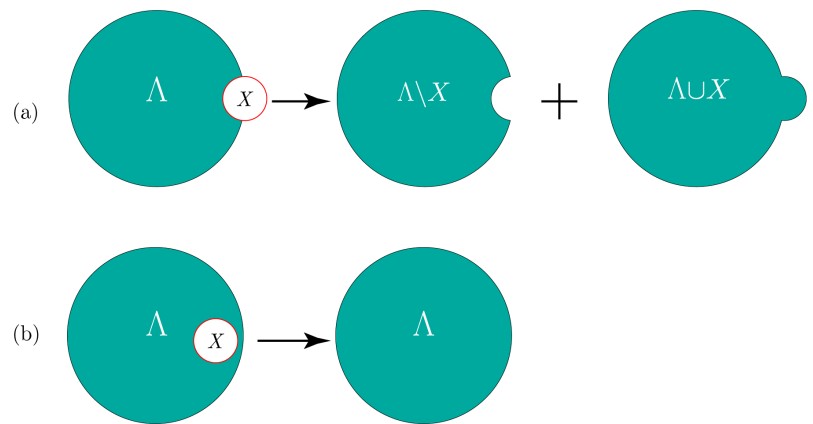

Figure 3: An illustration of the action of the superoperator $R_X$ on the swap operator $T_\Lambda$ with support on the region $\Lambda \subset V$. In $(a)$ the domain $X$ has a non trivial overlap with the boundary $\partial \Lambda$ and, according to Eq.(31), its action gives a linear combination of two domains, namely $\Lambda \setminus X$ and $\Lambda \cup X$. In $(b)$ the domain $X \subset \Lambda$ is completely contained in $\Lambda$ and its action is trivial. Note that we neglected the prefactors $N_{d_{\Lambda \setminus X}}$ and $N_{d_{\Lambda \cup X}}$, cfr. Eq.(31).

A simple representation of this action is provided in Fig.3. Note that we can compactly write the above action as follows:

$$R_X(T_\Lambda) = (1 - f(X, \Lambda))T_\Lambda + f(X, \Lambda)[N_{d_{\Lambda \cup X}} T_{\Lambda \cup X} + N_{d_{\Lambda \setminus X}} T_{\Lambda \setminus X}], \tag{32}$$

where $f : (X, \Lambda) \to \mathbb{R}$:

$$f(X, \Lambda) = \begin{cases} 1, & X \cap \partial \Lambda \neq \emptyset, X \cap \partial \bar{\Lambda} \neq \emptyset, \\ 0, & X \cap \partial \Lambda = \emptyset. \end{cases} \tag{33}$$

Eqs. (31) and (32) are telling us that if $X$ intersects the boundary of $\Lambda$, the expectation value of $T_\Lambda$ on $R_X(\sigma^{\otimes 2})$ becomes the linear combination of the expectation values on the original state $\sigma^{\otimes 2}$ of swap operators with different boundaries, namely $\Lambda \cup X$ and $\Lambda/X$. If $X$ is completely inside or outside $\Lambda$, the expectation value is unchanged. In other words, the action of $R_X$ on the swap $T_\Lambda$ results in a linear combination of two swaps defined in the patched regions $\Lambda \cup X$ and $\Lambda/X$. Thus, thanks to the duality in Eq.(30) we can write the expectation value of $R_X(T_\Lambda)$ on $\sigma^{\otimes 2}$ as a linear combination of the purity of $\sigma$ in different domains:

$$\langle R_X(T_\Lambda)\rangle_{\sigma^{\otimes 2}} = (1 - f(X, \Lambda))P_\Lambda(\sigma) + f(X, \Lambda)[N_{d_{\Lambda \cup X}} P_{\Lambda \cup X}(\sigma) + N_{d_{\Lambda \setminus X}} P_{\Lambda \setminus X}(\sigma)], \tag{34}$$

where $P_\Lambda(\sigma) = \langle T_\Lambda \rangle_{\sigma^{\otimes 2}}$ etc. Now, in order to generalize the above discussion to more than one domain $X$, consider the action of the quantum map $\mathcal{R}_S(\cdot)$ defined in Eq.(24). Although $\mathcal{R}_S(\cdot)$ is no more hermitian, we can exploit the duality as well and write:

$$\langle T_\Lambda \rangle_{\mathcal{R}_S(\sigma^{\otimes 2})} = \left\langle \mathcal{R}_S^\dagger(T_\Lambda) \right\rangle_{\sigma^{\otimes 2}}, \tag{35}$$

where $\mathcal{R}_S^\dagger(\cdot) = R_{\tilde{X}_1} \cdots R_{\tilde{X}_k}(\cdot)$. As one can see the adjoint operator $\mathcal{R}_S^\dagger(\cdot)$ is always the same operator $\mathcal{R}(\cdot)$ with a different ordering of the domains of $S$. Thus, defining the ordered subset

$$\bar{S} = \{\tilde{X}_k, \ldots, \tilde{X}_1 \,|\, \tilde{X}_i \in S\} \tag{36}$$

we can write $\mathcal{R}_S^\dagger(\cdot) = \mathcal{R}_{\bar{S}}(\cdot)$. Here *ordered* means that, given $\tilde{X}_i, \tilde{X}_j \in \bar{S}$ with $i > j$, the map $R_{\tilde{X}_j}(\cdot)$ acts after the map $R_{\tilde{X}_i}(\cdot)$. In order to avoid confusion, let us re-define the subsets as

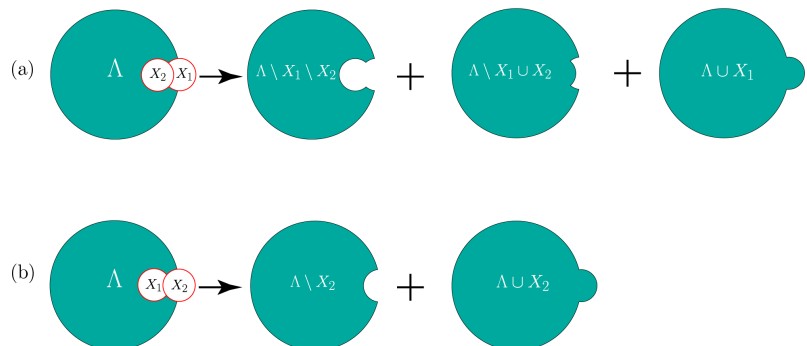

Figure 4: The figure shows how different orderings of the same domains $X_i \in \bar{S}$ can give different results. In $(a)$, $\bar{S}_{(a)} = \{X_1, X_2\}$ and thus we have $f(X_1, \Lambda) = 1$, $f(X_2, \Lambda \setminus X_1) = 1$, $f(X_2, \Lambda \cup X_1) = f(\Lambda, X_2) = 0$; therefore $\left\langle R_{\bar{S}_{(a)}}(T_\Lambda) \right\rangle_{\sigma^{\otimes 2}} = N_{\Lambda \setminus X_1} P_{\Lambda \cup X_1}(\sigma) + N_{\Lambda \setminus X_1} N_{d_{\Lambda \setminus X_1 \setminus X_2}} P_{\Lambda \setminus X_1 \setminus X_2}(\sigma) + N_{\Lambda \setminus X_1} N_{\Lambda \setminus X_1 \cup X_2} P_{\Lambda \setminus X_1 \cup X_2}(\sigma))$, cfr. Eqs. (33) and (38). In $(b)$, $\bar{S}_{(b)} = \{X_2, X_1\}$ and thus we have $f(X_1, \Lambda) = 0$, $f(X_2, \Lambda) = 1$; therefore $\left\langle R_{\bar{S}_{(b)}}(T_\Lambda) \right\rangle_{\sigma^{\otimes 2}} = N_{\Lambda \setminus X_2} P_{\Lambda \setminus X_2}(\sigma) + N_{\Lambda \cup X_2} P_{\Lambda \cup X_2}(\sigma)$.

$X_j = \tilde{X}_{k+1-j}$, so that $\bar{S} = \{X_1, \ldots, X_k \,|\, X_j = \tilde{X}_{k+1-j}, \tilde{X}_{k+1-j} \in S\}$ and $\mathcal{R}_{\bar{S}}(T_\Lambda) = R_{X_k} \cdots R_{X_1}(\cdot)$. By duality, the expectation value of $\mathcal{R}_S(T_\Lambda)$ is always a linear combination of purities of $\sigma$:

$$\langle T_\Lambda \rangle_{\mathcal{R}_S(\sigma^{\otimes 2})} = \left\langle \mathcal{R}_{\bar{S}}(T_\Lambda) \right\rangle_{\sigma^{\otimes 2}} = \sum_{\Lambda_\alpha \in \mathcal{Y}^{(k)}(\Lambda)} m_{\Lambda_\alpha} P_{\Lambda_\alpha}(\sigma), \qquad (37)$$

where we defined the set of domains $\mathcal{Y}^{(k)}(\Lambda) := \{\Lambda, \Lambda \cup X_1, \Lambda \cup X_2, \ldots, \Lambda \cup X_1 \setminus X_2, \ldots\}$; the set of domains $\mathcal{Y}^{(k)}(\Lambda)$ contains all the possible combinations of patching given by the domains (patches) $X_1, \ldots, X_k$; in the r.h.s of (37) the coefficients $m_{\Lambda_\alpha}$ depend on the particular choice of the ordered string $S$ and on the geometry of the region $\Lambda$. In order to make the notation clearer, let us write the expression for $k = 2$ explicitly:

$$
\begin{aligned}
\left\langle \mathcal{R}_{\bar{S}}(T_\Lambda) \right\rangle_{\sigma^{\otimes 2}} &\equiv \left\langle R_{X_2} R_{X_1}(T_\Lambda) \right\rangle_{\sigma^{\otimes 2}} = (1 - f(X_1, \Lambda))(1 - f(X_2, \Lambda)) P_\Lambda(\sigma) \\
&+ (1 - f(X_1, \Lambda)) f(X_2, \Lambda) [N_{d_{\Lambda \cup X_2}} P_{\Lambda \cup X_2}(\sigma) + N_{d_{\Lambda \setminus X_2}} P_{\Lambda \cup X_2}(\sigma)] \\
&+ f(X_1, \Lambda)(1 - f(X_2, \Lambda \cup X_1)) N_{d_{\Lambda \cup X_1}} P_{\Lambda \cup X_1}(\sigma) \\
&+ f(X_1, \Lambda)(1 - f(X_2, \Lambda \setminus X_1)) N_{d_{\Lambda \setminus X_1}} P_{\Lambda \setminus X_1}(\sigma) \\
&+ f(X_1, \Lambda) f(X_2, \Lambda \cup X_2)(N_{d_{\Lambda \cup X_1}} N_{d_{\Lambda \cup X_2}} P_{\Lambda \cup X_1 \cup X_2}(\sigma) + N_{d_{\Lambda \cup X_1}} N_{d_{\Lambda \setminus X_2}} P_{\Lambda \cup X_1 \setminus X_2}(\sigma)) \\
&+ f(X_1, \Lambda) f(X_2, \Lambda \setminus X_2)(N_{d_{\Lambda \setminus X_1}} N_{d_{\Lambda \cup X_2}} P_{\Lambda \setminus X_1 \cup X_2}(\sigma) + N_{d_{\Lambda \setminus X_1}} N_{d_{\Lambda \setminus X_2}} P_{\Lambda \setminus X_1 \setminus X_2}(\sigma)),
\end{aligned}
\qquad (38)
$$

where $\bar{S} = \{X_1, X_2\}$ and $m_\Lambda \equiv (1 - f(X_1, \Lambda))(1 - f(X_2, \Lambda))$, $m_{\Lambda \setminus X_1 \setminus X_2} \equiv N_{d_{\Lambda \setminus X_1}} N_{d_{\Lambda \setminus X_2}} f(X_1, \Lambda) \times f(X_2, \Lambda \setminus X_2)$, etc. The model is completely general: once one has chosen the string $S$ and the domain $\Lambda$ the functions $f$ are completely determined to be either 0 or 1 according to the rules in Eq.(33). It is worth noting that the ordering of the domains $X_i \in \bar{S}$ is very important; consider $X_1, X_2 \in \bar{S}$ and note it can be the case that $X_2 \cap \partial(\Lambda/X_1) \neq 0$ while $X_2 \cap \partial \Lambda = 0$ and so in the if $X_2$ acts before $X_1$ it does not have any effect, see Fig.4 for a pictorial proof.

## 3.3 Topology at a large scale: $l-$topology

After establishing the fact (i), we now see how the proof would work. A shallow map $\mathcal{R}$ will deform the boundary of the subsystems $A, B, C, D$ only locally, in a way that in the ratio Eq.(15) there is an exact cancellation for the boundary modifications.

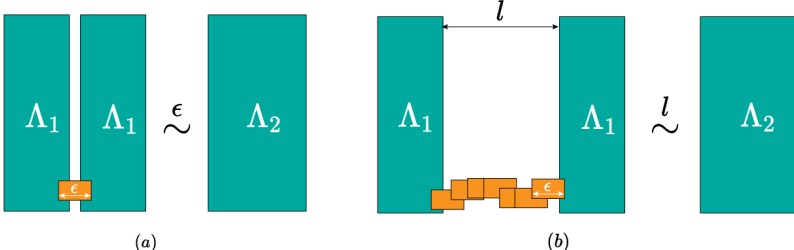

Figure 5: (a) Illustration of $\epsilon-$topology: $\Lambda_1$ becomes topologically equivalent to $\Lambda_2$ by a patch of diameter $\epsilon$, see Eq. (39). (b) Illustration of $l-$topology: in this case a combination of many $\epsilon-$patches is necessary to make $\Lambda_1$ topologically equivalent to $\Lambda_2$.

The boundary deformations induced by $\mathcal{R}$ are not strictly speaking topological, because they can punch holes or glue disconnected parts. However, they can only punch holes and glue parts on a short scale. For what we are concerned, only topological changes at a large scale are important. For this reason, in the following we introduce the notion of topological equivalence by "small scale patching".

Consider a graph $\Gamma = (V, \mathcal{E})$ and a subset of vertices (region) $\Lambda \subset V$; we define the complement of $\Lambda$ as $\bar{\Lambda} := \{x \in V \,|\, x \notin \Lambda\}$; then the (inner) *boundary* of $\Lambda$ is defined as $\partial \Lambda := \{x \in \Lambda \,|\, (x, y) \in E, y \in \bar{\Lambda}\}$. $\Lambda$ has two (or more) disconnected boundaries if $\partial \Lambda = \partial \Lambda_1 \cup \partial \Lambda_2$ and one of the following properties is satisfied:

- $\Lambda = \Lambda_1 \cup \Lambda_2$ and there is no path in $\Lambda$ connecting $\Lambda_1$ and $\Lambda_2$.

- $\bar{\Lambda} = \bar{\Lambda}_1 \cup \bar{\Lambda}_2$ and there is no path in $\bar{\Lambda}$ connecting $\bar{\Lambda}_1$ and $\bar{\Lambda}_2$.

The above definitions immediately generalize to more than two, say $n_\partial(\Lambda)$, disconnected boundaries. In the following the number of disconnected boundaries will be denoted as $n_\partial(\Lambda)$.

An $\epsilon-$*patch* is a simply connected region $X \subset V$ of diameter $\text{diam}(X) = \epsilon$. Considering a region $\Lambda \in V$, the *patching* of $\Lambda$ through $X$ is defined as the map:

$$\mathscr{P}_X(\Lambda) := \begin{cases} \Lambda \cup X, & \text{or} \\ \Lambda/X, \end{cases} \tag{39}$$

i.e. the action of patching whether adds something to the region $\Lambda$ or subtracts it. The composition of patching $A$ by $X_1$ and $X_2$ can result in one of the following four combinations $\Lambda \cup X_1 \cup X_2$, $\Lambda \cup X_1/X_2$, $\Lambda/X_1 \cup X_2$ and $\Lambda/X_1/X_2$. It is clear that by combining more $\epsilon-$patches together $X = X_1 \cup \cdots \cup X_{n_p}$ one can create a patch $X$ as large as one wants, resulting in $\Lambda \cup X$ or $\Lambda/X$. Having introduced the notion of patching a region $\Lambda \subset V$, we can introduce a notion of distance in the topology:

**Definition 2** ($\epsilon-$Topology). *Two regions $\Lambda_1, \Lambda_2 \subset V$ are $\epsilon-$topologically equivalent $\Lambda_1 \overset{\epsilon}{\sim} \Lambda_2$ iff $\mathscr{P}_X(\Lambda_1) \sim \Lambda_2$, i.e. patching $\Lambda_1$ with a $\epsilon-$patch makes $\Lambda_1$ topologically equivalent to $\Lambda_2$.*

The above definition naturally extends to a combination of more than one patch: two regions $\Lambda_1, \Lambda_2 \subset V$ are $l-$topologically equivalent $\Lambda_1 \overset{l}{\sim} \Lambda_2$ iff $\mathscr{P}_X(\Lambda_1) \sim \Lambda_2$ where $\text{diam}(X) \equiv \text{diam}(X_1 \cup \cdots \cup X_{n_p}) = l$. See 5 for a pictorial representation. At this point, we want to formalize the notion of how some sets can contain the same points and have the same shape locally, thus also having the same boundary length, and yet have a different number of disconnected boundaries. Consider the doubled graph $\Gamma^2 := (V^2, \mathcal{E}^2)$ and define the set:

$$E_{\Lambda_1 \Lambda_2} := \{\{x_1, x_2\} \,|\, x_1 \in \Lambda_1, x_2 \in \Lambda_2, \Lambda_1, \Lambda_2 \subset V\}, \tag{40}$$

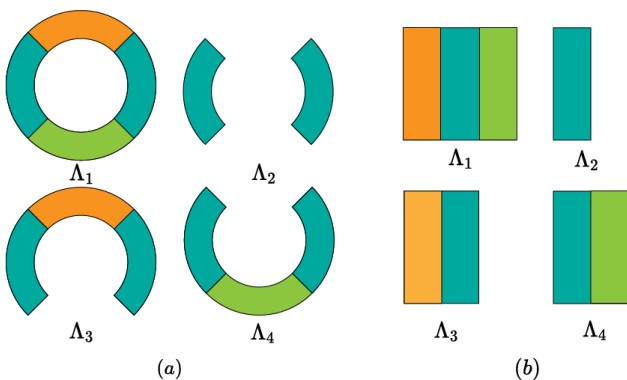

Figure 6: Pictorial representation of genuine topological difference. Consider the sets $\Lambda_1, \Lambda_2, \Lambda_3$ and $\Lambda_4$ sketched in both $(a)$ and $(b)$. In both $(a)$ and $(b)$ the sets $E_{\Lambda_1\Lambda_2}$ and $E_{\Lambda_1\Lambda_2}$ are equal $E_{\Lambda_1\Lambda_2} = E_{\Lambda_3\Lambda_4}$. While $(a)$ the sets show genuine topological difference $E_{\Lambda_1\Lambda_2} \not\sim E_{\Lambda_3\Lambda_4}$, because $n_\partial(E_{\Lambda_1\Lambda_2}) - n_\partial(E_{\Lambda_3\Lambda_4}) = 2$; $(b)$ the sets do not have any topological difference $E_{\Lambda_1\Lambda_2} \sim E_{\Lambda_3\Lambda_4}$, indeed $n_\partial(E_{\Lambda_1\Lambda_2}) - n_\partial(E_{\Lambda_3\Lambda_4}) = 0$.

where the elements $\{x_1, x_2\}$ are non-ordered pairs of elements of $\Lambda_1$ and $\Lambda_2$; we define the boundary $\partial E_{\Lambda_1\Lambda_2}$ of $E_{\Lambda_1\Lambda_2}$ as:

$$\partial E_{\Lambda_1\Lambda_2} := \{\{x_1, x_2\} \,|\, x_1 \in \partial\Lambda_1, x_2 \in \partial\Lambda_2\}, \tag{41}$$

where $|\partial E_{\Lambda_1\Lambda_2}| := |\partial\Lambda_1| + |\partial\Lambda_2|$ and the number of disconnected boundaries $n_\partial(E_{\Lambda_1\Lambda_2}) := n_\partial(\Lambda_1) + n_\partial(\Lambda_2)$. At this point, we require that, considering four regions $\Lambda_1, \Lambda_2, \Lambda_3, \Lambda_4$, the two sets obey $E_{\Lambda_1\Lambda_2} = E_{\Lambda_3\Lambda_4}$, that is: $E_{\Lambda_1\Lambda_2}$. Moreover, we require that $E_{\Lambda_3\Lambda_4}$ are equal as sets and their boundaries $\partial E_{\Lambda_1\Lambda_2}, \partial E_{\Lambda_3\Lambda_4}$ are equal (as sets) with equal lengths $|\partial E_{\Lambda_1\Lambda_2}| = |\partial E_{\Lambda_3\Lambda_4}|$, see Fig. 6, but they have a *genuine topological difference*, i.e. the number of disconnected boundaries within the two sets is different (Fig. 6 $(a)$), as the following definition says:

**Definition 3** (Genuine topological difference)**.** *Consider four regions $\Lambda_1, \Lambda_2, \Lambda_3, \Lambda_4$ and the two sets $E_{\Lambda_1\Lambda_2}, E_{\Lambda_3\Lambda_4}$ such that, $E_{\Lambda_1\Lambda_2} = E_{\Lambda_3\Lambda_4}$. If $n_\partial(E_{\Lambda_1\Lambda_2}) \neq n_\partial(E_{\Lambda_3\Lambda_4})$, the two sets enjoy a genuine topological difference, i.e. $E_{\Lambda_1\Lambda_2} \not\sim E_{\Lambda_3\Lambda_4}$.*

See Fig.6 for an illustration of the above definition.

With the notion of $l-$topology given in Definition 2 we can define:

**Definition 4** ($l-$genuine topological difference)**.** *Consider four regions $\Lambda_1, \Lambda_2, \Lambda_3, \Lambda_4$ and the two sets $E_{\Lambda_1\Lambda_2}$ and $E_{\Lambda_3\Lambda_4}$ such that $E_{\Lambda_1\Lambda_2} \not\sim E_{\Lambda_3\Lambda_4}$ and $n_\partial(E_{\Lambda_1\Lambda_2}) - n_\partial(E_{\Lambda_3\Lambda_4}) = t$. The two sets $E_{\Lambda_1\Lambda_2}$ and $E_{\Lambda_3\Lambda_4}$ shows $l-$genuine topological difference iff for any $l'-$patch such that $l' < l$ acting on $\Lambda_1, \Lambda_2, \Lambda_3, \Lambda_4$, the sets feature the same genuine topological difference, i.e.*

$$n_\partial(\mathscr{P}_X(E_{\Lambda_1\Lambda_2})) - n_\partial(\mathscr{P}_X(E_{\Lambda_3\Lambda_4})) = t, \tag{42}$$

*where $\mathscr{P}_X(E_{\Lambda_1\Lambda_2}) := \{\{x_1, x_2\} \,|\, x_1 \in \mathscr{P}_X(\Lambda_1), x_2 \in \mathscr{P}_X(\Lambda_2)\}$.*

*Example.—* A specific example of $l-$topologically different sets is provided in Fig. 7. We see that while by patching a region $\Lambda_1$ one can change its topology, on the other hand if the four sets $\Lambda_1, \Lambda_2, \Lambda_3, \Lambda_4$ shows genuine $l-$topological difference, no matter the patching on the four sets, the difference of disconnected boundaries will remain the same as long as the diameter of the patch fulfills the condition $l' < l$.

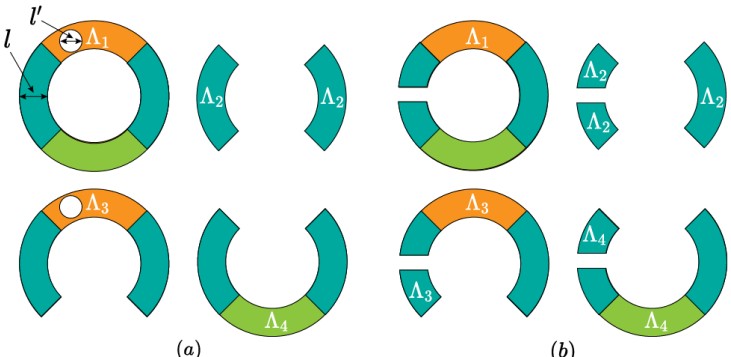

Figure 7: Pictorial representation of $l-$genuine topological difference. The typical size of the sketched domain is $l$. $(a)$ A $l'$ patch has been applied to the sets of domains $E_{\Lambda_1 \Lambda_2}$ and since $l' < l$ the sets shows the same genuine topological difference, indeed $n_\partial(E_{\Lambda_1 \Lambda_2}) = 3 + 2$, $n_\partial(E_{\Lambda_3 \Lambda_4}) = 2 + 1$ and $n_\partial(E_{\Lambda_1 \Lambda_2}) - n_\partial(E_{\Lambda_3 \Lambda_4}) = 2$, cfr. Fig. 6. $(b)$ the sets are patched with a patch whose size is $l' \geq l$ which change the genuine topological difference, indeed $n_\partial(E_{\Lambda_1 \Lambda_2}) = 1 + 3$, $n_\partial(E_{\Lambda_1 \Lambda_2}) = 2 + 2$ and $n_\partial(E_{\Lambda_1 \Lambda_2}) - n_\partial(E_{\Lambda_3 \Lambda_4}) = 0 \neq 2$.

## 3.4 Proof of the main result

We are finally ready to prove the main result of the paper. By virtue of fact (i), we can re-write Eq.(15) for the topological purity of the state $\Psi_S = \mathcal{R}_S(\Psi_0^{\otimes 2})$ in terms of expectation values of evolved swap operators $\mathcal{R}_{\bar{S}}(T_\Lambda)$ for $\Lambda$ being $AB, BC, B$ and $ABC$, namely:

$$\tilde{P}_{top}(\Psi_S) = \frac{\left\langle \mathcal{R}_{\bar{S}}(T_{AB}) \right\rangle_{\Psi_0^{\otimes 2}} \left\langle \mathcal{R}_{\bar{S}}(T_{BC}) \right\rangle_{\Psi_0^{\otimes 2}}}{\left\langle \mathcal{R}_{\bar{S}}(T_B) \right\rangle_{\Psi_0^{\otimes 2}} \left\langle \mathcal{R}_{\bar{S}}(T_{ABC}) \right\rangle_{\Psi_0^{\otimes 2}}} . \tag{43}$$

Let $\mathcal{S}$ be the set of all possible strings of domains defined in Eq. (26) and let $\tilde{\mathcal{S}}_l \subset \mathcal{S}$ be the subset containing all possible $l-$shallow strings, see Definition 1.

To prove the main theorem, we need to prove that, for any $S \in \tilde{\mathcal{S}}_l$, the topological purity keeps constant to the value of the topological purity of the initial state, i.e. the ratio of expectation values of evolved swap operators in Eq.(15) equals the ratio of expectation values of the initial $T_{AB}, T_{BC}, T_B, T_{ABC}$. We recall that, evolving a swap operator $T_\Lambda$ by $\mathcal{R}_S$, one deforms the domain $\Lambda$ by patching. In particular we consider two states, the ground state of the quantum double model and a topologically trivial state. If the initial state $\left| \Psi_{qd} \right\rangle \in \mathcal{L}$ is the ground state of the quantum double model (cfr. Sec.1 ) then $\tilde{P}_{top}(\Psi_{qd}^{\otimes 2}) = P_{top}(\Psi_{qd}) = 2^{-2\gamma}$, while if the initial state is a pure and topologically trivial state such as $\Psi_{triv}$, then $\tilde{P}_{top}(\Psi_{triv}^{\otimes 2}) = P_{top}(\Psi_{triv}^{\otimes 2}) = 1$.

*Proof.* Consider $\left| \Psi_{qd} \right\rangle \in \mathcal{L}$ and a shallow string $S \in \tilde{\mathcal{S}}_l$ defined in Definition 1. Let us compute the topological purity of $\Psi_S \equiv \mathcal{R}_S(\Psi_{qd}^{\otimes 2})$ for $S \in \tilde{\mathcal{S}}_l$. According to Eq.(43) we can directly compute the expectation values of the evolution of the swap operators $T_\Lambda$ for $\Lambda = (AB, BC, B, ABC)$. Recalling Eq.(37) the expectation value of the evolved swap operator $T_\Lambda$ is a linear combination of purities of $\Psi_{qd}$ in domains $\Lambda_\alpha \in \mathcal{Y}_k(\Lambda)$:

$$\left\langle \mathcal{R}_{\bar{S}}(T_\Lambda) \right\rangle_{\Psi_0^{\otimes 2}} = \sum_{\Lambda_\alpha \in \mathcal{Y}_k(\Lambda)} m_{\Lambda_\alpha} P_{\Lambda_\alpha}(\Psi_{qd}). \tag{44}$$

According to Eq. (8), any purity term $P_{\Lambda_\alpha}(\Psi_0)$, for $\Lambda_\alpha \in \mathcal{Y}_k(\Lambda)$ equals to

$$P_{\Lambda_\alpha}(\Psi_{qd}) = 2^{-\log_2 |d\partial \Lambda_\alpha| + \Gamma_{\Lambda_\alpha}} , \tag{45}$$

where we recall that $|\partial \Lambda_\alpha|$ is the boundary length of $\Lambda_\alpha$ and $\Gamma_{\Lambda_\alpha} = \gamma_\Lambda + n_\partial(\Lambda)\gamma$ is the sum of a geometrical term $\gamma_\Lambda$ and a pure topological term $n_\partial(\Lambda)\gamma$ only depending on the topological nature of the state and proportional to the number of disconnected boundaries $n_\partial(\Lambda)$ of the domain $\Lambda$. Plugging Eq. (44) in Eq.(43) one obtains

$$\tilde{P}_{top}(\Psi_S) = \frac{\sum_\alpha m_{AB_\alpha} P_{AB_\alpha} \sum_\beta m_{BC_\beta} P_{BC_\beta}}{\sum_\eta m_{B_\eta} P_{B_\eta} \sum_\zeta m_{ABC_\zeta} P_{ABC_\zeta}}, \tag{46}$$

where we adopted a compact notation for the sum, namely $\sum_\alpha \equiv \sum_{AB_\alpha \in \mathcal{Y}_k(AB)}$ and $P_{AB_\alpha} \equiv P_{AB_\alpha}(\Psi_{qd})$, etc.

Now, if the two sets $E_{(ABC)(B)}$ and $E_{(AB)(BC)}$ enjoy a genuine $l-$topological difference $E_{(ABC)(B)} \not\sim E_{(AB)(BC)}$, then the difference in the number of their boundaries remains unchanged under any patches of diameter less than $l$:

$$n_\partial(E_{(ABC)(B)}) - n_\partial(E_{(AB)(BC)}) = 2. \tag{47}$$

Let us rewrite Eq. (46) as

$$\tilde{P}_{top}(\Psi_S) = \frac{\sum_{\alpha,\beta} m_{AB_\alpha} m_{BC_\beta} P_{AB_\alpha} P_{BC_\beta}}{\sum_{\eta,\zeta} m_{B_\eta} m_{ABC_\zeta} P_{B_\eta} P_{ABC_\zeta}}. \tag{48}$$

As proven in Sec. 3.1, the action of the noisy map corresponding to the string $S = \{X_1, \ldots, X_k | X_i \subset V\}$ on a swap operator $T_\Lambda$ results in a combination of purity of the initial state $\Psi_0^{\otimes 2}$ computed in all the possible patched regions with $X_1, \ldots, X_k$ according to the rules in Eq. (37). Since we assumed that $S$ is a shallow string, we also have that the combination of patches can never creates a patch of diameter bigger than $l$ and thus the action of the noisy map keeps constant the genuine $l-$topological difference of the initial domains $ABC, B, AB, BC$. This property is reflected in the following: for any $\alpha, \beta$ the purity in the patched regions $AB_\alpha, BC_\beta$ can be written as

$$P_{AB_\alpha} P_{BC_\beta} = 2^{-(|\partial AB_\alpha| + |\partial BC_\beta|) + (\gamma_{AB_\alpha} + \gamma_{BC_\beta}) + \gamma(n_\partial(AB_\alpha) + n_\partial(BC_\beta))}, \tag{49}$$

then, there are two corresponding patched regions, $ABC_\eta$ and $B_\zeta$, such that $E_{(ABC_\eta)(B_\zeta)}$ and $E_{(AB_\alpha)(BC_\beta)}$ shows genuine topological difference $E_{(ABC_\eta)(B_\zeta)} \not\sim E_{(AB_\alpha)(BC_\beta)}$ and the number of disconnected boundaries obeys to:

$$n_\partial(E_{(ABC_\eta)(B_\zeta)}) - n_\partial(E_{(AB_\alpha)(BC_\beta)}) = 2, \tag{50}$$

thus we have $|\partial AB_\alpha| + |\partial BC_\beta| = |\partial ABC_\eta| + |\partial B_\zeta|$, $\gamma_{AB_\alpha} + \gamma_{BC_\beta} = \gamma_{ABC_\eta} + \gamma_{B_\zeta}$ and $n_\partial(AB_\alpha) + n_\partial(BC_\beta) = -2 + n_\partial(ABC_\eta) + n_\partial(B_\zeta)$. The product of purities in Eq. (49) is therefore equal to:

$$P_{AB_\alpha} P_{BC_\beta} = 2^{-2\gamma} P_{ABC_\eta} P_{B_\zeta}. \tag{51}$$

In order to conclude the proof, it is worth noting that the weights $m_\Lambda$ of purities, in Eq. (44), do not depend on the number of disconnected boundaries $n_\partial(\Lambda)$, cfr. Sec. 3.2. Then, because the sets $E_{(ABC_\eta)(B_\zeta)}$ and $E_{(AB_\alpha)(BC_\beta)}$ enjoy genuine topological difference, for any $\alpha$ and $\beta$ there exist $\eta$ and $\zeta$ such that we have the following:

$$m_{AB_\alpha} m_{BC_\beta} = m_{ABC_\eta} m_{B_\zeta}. \tag{52}$$

Note that we have the equality because the sets $E_{(ABC_\eta)(B_\zeta)} = E_{(AB_\alpha)(BC_\beta)}$ are equal. Finally, by grouping all the terms, we have:

$$\tilde{P}_{top}(\Psi_S) = 2^{-2\gamma} \frac{\sum_{\eta,\zeta} m_{B_\eta} m_{ABC_\zeta} P_{B_\eta} P_{ABC_\zeta}}{\sum_{\eta,\zeta} m_{B_\eta} m_{ABC_\zeta} P_{B_\eta} P_{ABC_\zeta}} = 2^{-2\gamma} = \tilde{P}_{top}(\Psi_{qd}^{\otimes 2}). \tag{53}$$

The proof for $\Psi_{triv}$ being a pure and topologically trivial state is identical to the one presented above, with the only difference that $P_{\Lambda_\alpha}(\Psi_{triv}) = \left\langle T_{\Lambda_\alpha} \right\rangle_{\Psi_{triv}^{\otimes 2}} = 1$ for any $\Lambda_\alpha \in \mathcal{Y}_k(\Lambda)$, cfr. Eq.(37). This concludes the proof. $\qquad\qquad\qquad\qquad\qquad\qquad\qquad\qquad\qquad\qquad\qquad\qquad\square$

**Remark.** *If the string $S \notin \tilde{\mathcal{S}}_l$ is not a shallow string, then $\mathrm{diam}(X_1 \cup \ldots X_k) > l$ and it can be the case that the joint patch creates a hole in the donut shape of ABC or connects two far apart regions (operations breaking the genuine topological difference), see Fig.7 (b) for a graphical example. In that case, there exist $\bar{\alpha}, \bar{\beta}, \bar{\eta}$ and $\bar{\zeta}$ in Eq. (48) such that $n_\partial(E_{(AB_{\bar{\alpha}})(BC_{\bar{\beta}})}) - n_\partial(E_{(ABC_{\bar{\eta}})(B_{\bar{\zeta}})}) \neq 2$ which would invalidate Eq. (51) for $P_{AB_{\bar{\alpha}}}, P_{BC_{\bar{\beta}}}, P_{ABC_{\bar{\eta}}}$ and $P_{B_{\bar{\zeta}}}$; that would result in the failing of the grouping in Eq. (53), hence $P_{top}(\Psi_S) \neq 2^{-2\gamma}$.*

## 4 Conclusions

In this paper, we addressed some questions regarding the stability of topological order under noisy perturbations, but the path to find a general analytic proof is still long and tortuous. Working with the ground state $\Psi_0$ of quantum double models, we defined a new probe for topological order - the topological purity $\tilde{P}_{top}$ - proving its robustness in two distinct phases, namely the topological phase and the trivial phase. More precisely, as a noise model, we introduced a set of quantum maps that mimics the evolution of local random quantum circuits. The two phases are indeed created by the quantum maps as orbits of two initially distinct states, the ground state of quantum double models $\Psi_0$ and a pure, topologically trivial state. We found that the topological purity attains two different constant values among such states, in particular $\tilde{P}_{top} = 2^{-2\gamma} < 1$ for the topologically ordered phase and $\tilde{P}_{top} = 1$ for the trivial phase. The dynamics of the topological purity under such noise model can be mapped onto dynamics for the subsystems used to define the topological purity, cfr. Sec.2. This property enabled us to prove our main theorem and to provide many pictorial representations, giving the reader more intuition on the effects of the noisy dynamics. The noisy dynamics is based on local quantum circuits. As the depth of the circuits increases, the noise propagates and eventually, for a circuits scaling with the size of the topologically relevant scale, the topological phase breaks down. Our work shares the model of propagation of local perturbations studied in [32].

Despite the generality of the setup, in the sense that the noise model does not obey any particular symmetry or fine tuned feature, our result is not the final word regarding the stability of topological order, and even more general and complete proofs are necessary to go further in this direction. This paper opens a series of different questions that might be interesting to investigate, for instance, whether the proof can be extended to non-abelian quantum double models, the difficulty being that for non abelian groups the order of the group is in general hard to compute. Moreover, an important open problem is whether the higher moments of the purity under random quantum circuits obey some algebra that can be cast in the form of evolution of geometries. This would open the way to, on the one hand, compute generic Rényi entropies for the evolved states and from there the von Neumann entropy, on the other hand, associating evolutions under a quantum map to evolution of geometries would be a very useful tool for the study of topological phases away from equilibrium. Moreover, studying higher moments of the purity would also result in evaluating the fluctuations of the ratio and products of purity, thereby making the scope of our results valid for more general noise models. In particular, it would be interesting to study the application of random local quantum circuits to the theory of gapped domain walls between topologically ordered systems developed in [53–55].

## Acknowledgments

We acknowledge support from NSF award number 2014000.

## A Action of $R_X(\cdot)$ on the swap operator

In this appendix, we review the calculations given in [52] to obtain Eq.(31). First, we recall the definition of the quantum map $R_X(\cdot)$ given in Eq.(22).

$$R_X(\cdot) = \int d\mu(U|X)(U_X)^{\otimes 2}(\cdot)(U_X)^{\dagger \otimes 2}, \tag{54}$$

where $U_X$ is a unitary operator acting on $\mathcal{H}_X$. The action of $R_X(\cdot)$ over a swap operator $T_\Lambda$ is:

$$
\begin{aligned}
R_X(T_\Lambda) &= \int d\mu(U|X)(U_X)^{\otimes 2}T_\Lambda(U_X)^{\dagger \otimes 2} \\
&= \frac{\text{tr}_X(T_\Lambda(\mathbb{1}_X + T_X))}{2d_X(d_X+1)}(\mathbb{1}+T_X) + \frac{\text{tr}_X(T_\Lambda(\mathbb{1}_X - T_X))}{2d_X(d_X-1)}(\mathbb{1}-T_X),
\end{aligned}
\tag{55}
$$

where we made use of the Haar average techniques [56, 57] to compute the integral. Before proceeding, it is important to make a remark on the role of the domains in this calculation, we have to distinguish between two cases: the first one where $X \subseteq \Lambda$ or $X \nsubseteq \Lambda$, and the second one where $X \cap \Lambda \neq \emptyset$ and $X \cap \overline{\Lambda} \neq \emptyset$. For the first case, when $X \subseteq \Lambda$, the Eq.(55) becomes

$$
\begin{aligned}
R_X(T_\Lambda) &= T_{\Lambda \setminus X} \frac{\text{tr}_X(T_X(\mathbb{1}_X + T_X))}{2d_X(d_X+1)}(\mathbb{1}+T_X) + T_{\Lambda \setminus X}\frac{\text{tr}_X(T_X(\mathbb{1}_X - T_X))}{2d_X(d_X-1)}(\mathbb{1}_X - T_X) \\
&= \frac{1}{2}(T_{\Lambda \setminus X}(\mathbb{1}_X + T_X) - T_{\Lambda \setminus X}(\mathbb{1}_X - T_X)) = T_{\Lambda \setminus X}T_X = T_\Lambda,
\end{aligned}
\tag{56}
$$

where we used that the swap operator $T_\Lambda = T_{\Lambda \setminus X}T_X$, while if $X \nsubseteq \Lambda$ we obtain:

$$R_X(T_\Lambda) = \frac{1}{2}T_\Lambda(\mathbb{1}_X + T_X) + \frac{1}{2}T_\Lambda(\mathbb{1}_X - T_X) = T_\Lambda. \tag{57}$$

When instead $X \cap \Lambda \neq \emptyset$ and $X \cap \overline{\Lambda} \neq \emptyset$, we obtain:

$$
\begin{aligned}
R_X(T_\Lambda) &= T_{\Lambda \setminus X} \frac{\text{tr}_X(T_{\Lambda \cap X}(\mathbb{1}_X + T_X))}{2d_X(d_X+1)}(\mathbb{1}+T_X) + T_{\Lambda \setminus X}\frac{\text{tr}_X(T_{\Lambda \cap X}(\mathbb{1}_X - T_X))}{2d_X(d_X-1)}(\mathbb{1}_X - T_X) \\
&= \frac{d_X^2 d_{\Lambda \cap X}^{-1} + d_{\Lambda \cap X}d_X}{2d_X(d_X+1)}(T_{\Lambda \setminus X} + T_{\Lambda \setminus}T_X) - \frac{d_X^2 d_{\Lambda \cap X}^{-1} - d_{\Lambda \cap X}d_X}{2d_X(d_X-1)}(T_{\Lambda \setminus X} - T_{\Lambda \setminus X}T_X) \\
&= N_{d_{\Lambda \setminus X}}T_{\Lambda \setminus X} + N_{d_{\Lambda \cup X}}T_{\Lambda \cup X},
\end{aligned}
\tag{58}
$$

where we used that $T_X = T_{X/(\Lambda \cap X)}T_{\Lambda \cap X}$ and that $T_{\Lambda \cup X} = T_{\Lambda \setminus X}T_X$ $N_{d \setminus X} = (d_X^2)$, with $N_{d_{\Lambda \setminus X}} := (d_X^2 - d_{\Lambda \cap X}^2)d_{\Lambda \cap X}^{-1}/(d_X^2 - 1)$ and $N_{d_{\Lambda \cup X}} := d_X(d_{\Lambda \cap X}^2 - 1)d_{\Lambda \cap X}^{-1}/(d_X^2 - 1)$ and $d_{\Lambda \cap X} = \dim \mathcal{H}_{\Lambda \cap X}$.

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
