# Peer review of "Stability of topological purity under random local unitaries"

_SciPost Physics, doi:SciPost Phys. 12, 096 (2022)_

## Round 1 · Referee Report · Anonymous (Referee 1) · 2022-1-4

Strengths

Rigorous derivation for the main result.

Weaknesses

No obvious weakness. See the report below for minor suggestions and comments.

Report

While topological entanglement entropy has been widely applied to characterize topologically ordered phases of matter, its stability throughout the entire phase lacks rigorous, analytical proof as pointed out by the authors. Motived by this observation, the authors introduce a quantity dubbed "topological purity", which captures the same pattern of long-range entanglement as topological entanglement entropy. The authors are able to prove a rigorous statement for the robustness of the topological purity under a noise model consisting of a shallow circuit with random local unitaries in the context of the quantum double model.

The proof relies on the following two non-trivial observations: (i) the purity of a subsystem in the ground state of the quantum double model only depends on the geometry of the subsystem boundary, and (ii) the random local perturbation is equivalent to deforming the subsystem boundary without changing the state in the calculation of subsystem purity based on the swap operator.

Since it is in general challenging to derive an exact, rigorous statement that is non-trivial regarding the stability of quantum phases of matter, I would recommend the publication, as long as the authors properly address the following comments of mine.

Some remarks/comments:

(1) The authors cited Ref.25-29 when reviewing the past numerical works that demonstrate the robustness of topological entanglement entropy under local perturbations. However, it is incorrect to cite Ref.25 as the topological entanglement entropy was not studied there.

(2) If I recall correctly, properly defining topological entanglement entropy based on the setup in Fig.2(a) requires both $l$ and $r$ scaling linearly with the system size L to avoid the effect of finite correlation length in a topologically ordered phase. Shouldn’t it mean that $l$ and $r$ scale with $O( \sqrt{N} )$ instead of $O(N^2)$ in Fig.2 caption? $N$ labels the number of qudits defined on links based on the notation defined on Page 2 and 3.

(3) Related to the above question, the main theorem on Page 9 basically states that if a string is $l$-shallow, then the topological purity is invariant under the corresponding application of random perturbation given by the string. Should we impose the condition that the string is $r$-shallow as well ($r$ is defined in Fig.2(a))?

(4) The topological purity is basically the exponential of the conventional topological 2-Renyi entropy defined as $S_{2,topo} = S_2(\rho_{ABC}) +S_2(\rho_{B}) -S_2(\rho_{AB}) - S_2(\rho_{BC})$, where $S_2(\rho_{\Lambda} ) = - \log ( \text{tr}_{\Lambda} \rho_{\Lambda}^2 )$ with $\Lambda$ denoting the chosen subregion. In the noise model considered, the authors essentially consider computing $- \log ( \text{tr}_\Lambda \overline{\rho_{\Lambda}^2 } )$, namely, the average of the purity instead of $- \overline{ \log ( \text{tr}_\Lambda \rho_{\Lambda}^2 ) }$, which would result in the average of topological entanglement entropy. In other words, while topological entropy and topological purity capture the same physics for a given state, their behavior after averaging over the shallow random circuits could in principle be different. I think authors should make it clear regarding the distinction between these two quantities in the draft.

In addition, the main statement proved by the authors in this work is actually weaker than one should expect for topological purity (or topological entanglement entropy). Namely, we expect the topological invariant (either topological purity or topological entropy) should remain constant under any single instance of a random shallow circuit. However, the authors only prove that the aforementioned topological quantity remains a constant "on average", which is therefore a weaker statement. It would be great if the authors can make this caveat clear in the manuscript. On the other hand, I was wondering whether it is possible to quantify the fluctuation/randomness of topological purity under the noise model considered by the authors. If one can show that the fluctuation is vanishing, then the statement regarding the average (studied in this work) would apply to any single instance of the shallow circuit.

Typos:

(1) In the paragraph right after Eq.51: "...do not depend on the number on the number of..." -> "...do not depend on the number of...".

(2) “Von Neumann” -> “von Neumann” throughout the draft.

(3) Page 5: “which is a entropic quantity” -> “which is an entropy quantity”

Requested changes

The authors should fix the typos and address my comments listed in the report above.

  • validity: good
  • significance: high
  • originality: good
  • clarity: high
  • formatting: good
  • grammar: good

Author:  Salvatore Francesco Emanuele Oliviero  on 2022-01-08  [id 2080]

(in reply to Report 1 on 2022-01-04)

We are grateful to the referee for their attentive readings and clarification requests.

  1. As pointed out by the referee reference 25 does not refer to the topological entanglement entropy, so we removed reference 25 from our references.
  2. We thank the referee for noticing the error in the caption of Fig. 2(a): we are considering both $r,l\sim O(N)$, while $N^2$ is the total number of qudits. The caption of Fig. $2(a)$ now reads "$(a)$ The graph configuration to define the topological entropy, i.e. $I(A;C|B)$. $l$ is the feature size of this graph configuration and $l\sim O(N)$, where $N^2$ is the total number of qudits, while $r\sim O(N)$ is radius of this topologically non trivial domain. Here and throughout the paper we assume $r>l$."
  3. Referee is right, the string of domains must be both $l-$shallow and $r-$shallow. However, having set $r>l$, a $l-$shallow string is automatically an $r-$shallow string. For sake of completeness, we added the following line to the manuscript: "Based on Fig. 2, we remark that an $l-$shallow string is an $r-$shallow string as well, being $r>l$."
  4. We thank the referee for having pointed out this insightful aspect. To make our statement clearer, we added the following sentence to the introduction, which has gained completeness and clarity. "Let us make a remark for the scope of clarity. In principle, one could define the topological purity by considering the ratios of the purities first, and then averaging over the unitaries in the quantum map. This quantity would be equivalent to the one we defined only if the ensemble fluctuations were proven to be very small, which seems a formidable problem. Our definition goes around this problem because it nevertheless defines one quantity that has a constant value in the ensemble of states connected to the topologically ordered state and a different constant value in the trivial ensemble. The value of the definition is in its properties, that is, in being able to distinguish the two ensembles of states."
  5. We fixed the typo as suggested.

---

## Round 1 · Referee Report · Anonymous (Referee 2) · 2022-1-7

Strengths

1-It aims at a hard and meaningful problem.
2-The main result is the theorem on page 9. It is proved rigorously, as far as I am able to verify. (Yes, I took time to verify the proof.)
3-The figures are very helpful.

Weaknesses

1-The theorem proved by the author is weaker than the harder and more important problem the authors aimed to solve. That being said, I think the merit of this work is that it develops a unique view of this problem or at least some aspects of it.
2-In three places of the draft, the authors used words to describe the results, which are stronger than what the theorem says. This may confuse some readers. See the report for constructive suggestions.

Report

I am likely to recommend this article for publication after the authors consider the comments and discussions below. I think this work is creative in its way of simplifying the original (hard) problem into a more trackable problem so that partial progress can be made.

The article aims to provide a mathematical understanding of the stability of topological entanglement entropy of the ground states of topologically ordered systems under noise. Here the noise, in a broad sense, can be any local perturbation of the Hamiltonian. It is generally believed that this is reasonable to simplify the problem to the study of the stability under finite-depth quantum circuits.

This problem is important and known to be a hard problem. The topological entanglement entropy of the ground state is believed to be able to characterize the gapped phase, which roughly speaking is the quantum many-body system at large sizes. A gapped phase of matter (without symmetry) is robust under local perturbations; therefore, it is desirable to show that the topological entanglement entropy takes the same value throughout the phase. (The problem is known to be trickier due to exotic counterexamples, first described by Bravyi, so the correct problem should be that the chance to have counterexamples goes to zero in some yet to be precisely defined sense.) The difficulty of this problem is further due to the interacting nature of the system.

The authors study a simpler problem that is related to the original problem. They establish the invariant of a statistical average of the topology purity - a Rényi version of topological entanglement entropy for Abelian quantum double models. One key concept the authors introduce is "l-shallow string", which allows the authors to state and then prove the theorem. The notion of "l-shallow string" is a creative choice (of a special type of finite-depth circuit). This is interesting because this simplification and the Haar average make the problem tractable.

Requested changes

1-The theorem the authors prove is written on Page 9. In the statement, it refers to Definition 4, "l-shallow string". In the present draft, Definition 4 is on page 14; I suggest moving it just below the theorem. The reason is that Definition 4 is simple and does not depend on things between Page 9 and 14. I agree that the proof of the theorem needs some preparation and the current organization works great.

2-I think "l-shallow string" is a very useful concept. Moreover, according to my understanding, a depth-l quantum circuit may not be an l-shallow string. (Do I understand this point correctly?) It would be nice if the authors can emphasize this point.

3-The authors used the words "phase" (meaning phase of matter) correctly in most places. However, in three places, the usage is different from the known definition in literature. This can confuse people.

The first place is at the bottom of page 2: "We find that the topological purity distinguishes two phases of states, attaining two different constant values.

The second place is Page 9: " Since we found that the TP gets a constant value in two distinct ensembles of states, we claim that the topological purity is stable in the topologically ordered phase and in the topologically trivial phase."

The third place is the first paragraph of the Conclusions (page 16).

4-The authors wrote, "Unlike the Von Neumann entanglement entropy (whose measurement requires a complete state tomography of the
system), the 2−Rényi entropy is directly related to the purity which is an observable and can be measured directly". I totally agree with this observation. On the other hand, it is known that von Neumann entropy has a special nice property called strong subadditivity, namely $S_{AB}+S_{BC}-S_B-S_{ABC}\ge0$. It has been a source of beautiful theoretical insights. Especially that some progress has been made recently in understanding gapped phases of matter from states saturating the strong subadditivity. It would be very nice and balanced if the authors can mention this aspect of von Neumann entropy, i.e., $I(A:C|B)\ge 0$ for von Neumann entropy.

Minor things:
Figure 5 looks great. Therein "Topology"-> "topology".

  • validity: high
  • significance: good
  • originality: high
  • clarity: high
  • formatting: excellent
  • grammar: excellent

Author:  Salvatore Francesco Emanuele Oliviero  on 2022-01-08  [id 2081]

(in reply to Report 2 on 2022-01-07)

We thank the referee for their extremely careful reading of the manuscript.

  1. As suggested by the referee we moved definition 4 above the main theorem.
  2. Before Definition 2 we added the following line to make the concept clearer. "In the following definition, we give the notion of $l-$ shallow string, which is produced by the action of a shallow quantum circuit (although the opposite is not true, i.e. a shallow string can be also generated by a non-shallow circuit):"
  3. Referee is right in that we used the word `phase' in a somewhat confusing way in the places the referee indicated. A phase of the matter is an ensemble of states that are considered equivalent: for example, they enjoy the same symmetry (this criterion is not useful for topologically ordered states), or they are adiabatically connected, or they are connected by a shallow quantum circuit[48]. The kind of order that is revealed by this equivalence relation consists in the property that is conserved. For symmetry breaking states, the ordered phase is the phase of all the states that break the symmetry of the Hamiltonian in the same way (that is, they enjoy the same residual symmetry). In this context, the topological phase is revealed by the conservation of the topological purity. We added the previous explanation in the introduction for sake of clarity and changed the text in the three indicated places in the following way:
    1. At page 2 we modified the manuscript as follows: "We find that the topological purity attains two different constant values in the two ensembles of states obtained by acting with the shallow quantum maps on two reference states, one topologically ordered and the other one topologically trivial, thereby defining two phases."
    2. Because of the added text in the introduction, the passage "Since we found that the TP gets a constant value in two distinct ensembles of states, we claim that the topological purity is stable in the topologically ordered phase and in the topologically trivial phase." on page 9 and the first paragraph of the conclusions should now be clear.
  4. Following the referee suggestion, we added the following passage: "The topological entropy is defined as
    $$ S_{top}(\sigma)=S_{ABC}(\sigma)+S_{B}(\sigma)-S_{AB}(\sigma)-S_{BC}(\sigma)\equiv -I(A;C|B) $$
    where $S_{\Lambda}(\sigma)$ labels the von Neumann entropy of $\text{tr}_{\bar{\Lambda}}(\sigma)$ where $\bar{\Lambda}$ is the complement of $\Lambda$ with respect to $ABCD$. The von Neumann entropy has several important theoretical properties, specifically strong subadditivity[50]. In the context of topological entanglement, this results in $I(A;C|B)\ge 0$. On the other hand, the $2-$R\'enyi entropy is also a good measure of entanglement and it has the advantage of being more amenable for our scope. Moreover, this quantity can be measured without resorting to complete state tomography[37-39]."
  5. As suggested by the referee we fixed the typos near Figure 5

References [37] S. J. van Enk and C. W. J. Beenakker, Measuring $\operatorname{Tr}\rho$ non single copies of $\rho$ using random measurements,Physical Review Letters 108, 110503 (2012),doi:10.1103/PhysRevLett.108.110503

[38] D. A. Abanin and E. Demler, Measuring entanglement entropy of a generic many-body system with a quantum switch, Physical Review Letters 109, 020504 (2012), doi:10.1103/PhysRevLett.109.020504.

[39] A. J. Daley, H. Pichler, J. Schachenmayer and P. Zoller, : Measuring entanglement growth in quench dynamics of bosons in an optical lattice , Physical Review Letters 109, 020505 (2012), doi:10.1103/PhysRevLett.109.020505

[48] M. B. Hastings and X.-G. Wen, Quasiadiabatic continuation of quantum states: The stability of topological ground-state degeneracy and emergent gauge invariance, Physical Review B 72, 045141 (2005), doi:10.1103/PhysRevB.72.045141

[50] M. A. Nielsen and I. L. Chuang, Quantum information theory, p. 528–607, Cambridge University Press, doi:10.1017/CBO9780511976667.016 (2010)

---

## Round 1 · Referee Report · Anonymous (Referee 3) · 2022-2-6

Strengths

  1. The work addresses the important problem of rigorously proving that topological entropy quantities are generic throughout a stable zero temperature quantum phase of matter on the lattice.

  2. It involves some interesting proof ideas.

Weaknesses

The result only applies to topological purity and random unitary circuits, rather than the case of more direct interest based on the topological entanglement entropy and local perturbations to the Hamiltonian.

Report

The topological entanglement entropy and related quantities including the topological purity considered here are widely used theoretical and numerical tools for the identification of topological phases of matter. A topic of much recent interest due to their use as quantum codes.

This work presents a proof that one such quantity, the topological purity, is indeed generic under random local unitary circuits. While it would be nice to see this sort of result extended to the stability of topological entanglement entropy under generic local Hamilton perturbations, the current work is already substantial and contains some nice proof ideas.

I believe the work is suitable for SciPost and I recommend its publication.

Requested changes

  1. I am curious whether the authors considered the relation of their work to the entanglement bootstrap program in https://arxiv.org/abs/1906.09376 and subsequent works, i.e. do we expect generic states (under random local unitary circuit) within a quantum double phase to satisfy the assumptions of this work, or similarly the assumptions of the proof in Ref.[33]?

  2. typo: "although the proof works fairly good for quantum double models" good -> well

  • validity: top
  • significance: good
  • originality: high
  • clarity: high
  • formatting: excellent
  • grammar: good

Author:  Salvatore Francesco Emanuele Oliviero  on 2022-02-08  [id 2171]

(in reply to Report 3 on 2022-02-06)

We appreciate the positive assessment of our paper.

1.Referee indicated some work that is relevant to the context in which our paper is placed and it offers perspectives for future work. We have added references to the work on gapped domain walls in the conclusions as part of the outlook for future research. These are the modifications done to the manuscript:

The noisy dynamics is based on local quantum circuits. As the depth of the circuits increases, the noise propagates and eventually, for a circuits scaling with the size of the topologically relevant scale, the topological phase breaks down. Our work shares the model of propagation of local perturbations studied in[32].

In particular, it would be interesting to study the application of random local quantum circuits to the theory of gapped domain walls between topologically ordered systems developed in[53-55].

2.We fixed the typo in the manuscript

References

[32] I. H. Kim,Perturbative analysis of topological entanglement entropy from conditional independence, Physical Review B86, 245116 (2012), doi:10.1103/PhysRevB.86.245116 [53] B. Shi, K. Kato and I. H. Kim,Fusion rules from entanglement, Annals of Physics 418,168164 (2020), doi:https://doi.org/10.1016/j.aop.2020.168164 [54] B. Shi and I. H. Kim,Domain wall topological entanglement entropy, Physical Review Letters 126, 141602 (2021), doi:10.1103/PhysRevLett.126.141602 [55] B. Shi and I. H. Kim,Entanglement bootstrap approach for gapped domain walls, Physical Review B103, 115150 (2021), doi:10.1103/PhysRevB.103.115150.

---

## Editorial Decision

published